# Stress Urinary Incontinence: An Unsolved Clinical Challenge

**DOI:** 10.3390/biomedicines11092486

**Published:** 2023-09-07

**Authors:** Niklas Harland, Simon Walz, Daniel Eberli, Florian A. Schmid, Wilhelm K. Aicher, Arnulf Stenzl, Bastian Amend

**Affiliations:** 1Department of Urology, University of Tuebingen Hospital, 72076 Tuebingen, Germany; niklas.harland@med.uni-tuebingen.de (N.H.); simon.walz@med.uni-tuebingen.de (S.W.); urologie@med.uni-tuebingen.de (A.S.); 2Department of Urology, University Hospital Zurich, 8091 Zurich, Switzerland; daniel.eberli@usz.ch (D.E.); florian.schmid@usz.ch (F.A.S.); 3Centre for Medical Research, University of Tuebingen Hospital, Eberhard Karls University Tuebingen, 72076 Tuebingen, Germany; aicher@uni-tuebingen.de

**Keywords:** stress urinary incontinence (SUI), urodynamics, urethral hypermobility, intrinsic sphincter deficiency, pharmacological intervention, pelvic floor muscle training (PFMT), bulking agents, tension-free vaginal tape (TVT), male sling, artificial urinary sphincter

## Abstract

Stress urinary incontinence is still a frequent problem for women and men, which leads to pronounced impairment of the quality of life and withdrawal from the social environment. Modern diagnostics and therapy improved the situation for individuals affected. But there are still limits, including the correct diagnosis of incontinence and its pathophysiology, as well as the therapeutic algorithms. In most cases, patients are treated with a first-line regimen of drugs, possibly in combination with specific exercises and electrophysiological stimulation. When conservative options are exhausted, minimally invasive surgical therapies are indicated. However, standard surgeries, especially the application of implants, do not pursue any causal therapy. Non-absorbable meshes and ligaments have fallen into disrepute due to complications. In numerous countries, classic techniques such as colposuspension have been revived to avoid implants. Except for tapes in the treatment of stress urinary incontinence in women, the literature on randomized controlled studies is insufficient. This review provides an update on pharmacological and surgical treatment options for stress urinary incontinence; it highlights limitations and formulates wishes for the future from a clinical perspective.

## 1. Introduction

Urinary incontinence is a profound problem in everyday life for those affected and often leads to social withdrawal and massive restrictions in various areas of life. The incidence and prevalence are often only vaguely understood, as urinary incontinence remains a major taboo subject despite our open society. Broome reported a prevalence of between 15% and 35% of the adult population in 2003 [1]. Thus, it becomes evident that the problem of urinary incontinence is a major challenge for the health system. This concerns not only the necessary capacity to adequately treat this number of patients but also requires corresponding financial resources.

Urinary incontinence (UI) can be broken down into different forms, of which stress urinary incontinence (SUI) and urge urinary incontinence (UUI) are the most common entities. Whereas SUI dominates in women, UUI is the focus in men [1,2]. For SUI, intrinsic and extrinsic incontinence are discussed as they offer different options for therapy [3]. The prerequisite for diagnosis and treatment of incontinence is the courage of patients affected to present themselves to a doctor. Regrettably, in everyday life, many patients lack adequate supplies of pads and diapers. A structured and successful treatment of incontinent patients is made possible by numerous guidelines [2,4], including the guidelines of the European Association for Urology, which is constantly updated [5,6,7]. Consistently, primarily conservative therapies are recommended before interventional therapies are suggested.

Despite standardization and recommendation guidelines, nowadays, urinary incontinence is still not causally curable [8]. With regard to stress urinary incontinence, currently, only pelvic floor exercises can provide the advantage of targeted and long-term improvement of muscle function. This therapy yields mild to moderate success with its causal approach. Most of the interventional therapy options for SUI can lead to an improved function of the sphincter, and therefore, relevant symptomatic improvement. But they do not directly improve the sphincter muscle fibers, nor its vascularisation or enervation. In addition, numerous procedures rely on the implantation of prosthetic materials in patients. As shown below, this is often associated with far-reaching complications for those affected, and this has led to a paradigm shift in the last decade with regard to the use of meshes and tapes in prolapse and incontinence surgery [9]. This review article reflects the current standards of clinical diagnostics and therapy of SUI and provides insight into the current limitations of the options available.

## 2. Definition and Epidemiology of Stress Urinary Incontinence

Urinary incontinence is defined as the involuntary loss of urine, which is perceived as a social and hygienic problem and which can be clinically objectified [2]. SUI is characterized by urine loss during physical activity and can be divided into three degrees of severity: Grade 1 with urine loss when coughing, sneezing, and laughing, Grade 2 with urine loss when getting up, when walking, or under physical activity, and Grade 3 with urine loss while lying [2,10]. Risk factors for SUI differ between the sexes. Whereas pregnancy, vaginal delivery, and age are the main risk factors in women, SUI in men is less common and often associated with surgeries in the lower urinary tract [1,11].

Incidence and prevalence of UI are reported very differently [12]. Current reports assume that 10 million people are affected in Germany [13]. For adult women, a prevalence of UI of 23% in Spain, 44% in France, 41% in Germany, and 42% in the UK was reported [14,15]. In Norway, a prevalence of 25% among women, with 7% reporting moderate to severe UI, was seen [16]. In Canada, the prevalence of incontinence in women was of 28.8%, with a 68% share of SUI, while in men the overall UI prevalence was only 5.4%, with a 27% share of SUI reported [17]. Comparable figures were found in the USA [18].

The high prevalence is associated with high expenditure in the health system due to medical consultation and supply of incontinence materials. However, due to low social acceptance of the disease, numerous patients bear the financial burden of incontinence pads themselves, and only 25% to 40% of women affected consulted a doctor to improve their condition and less than 5% received surgery to treat SUI [14,19]. An average demand of $ 750 per year for managing incontinence for an individual woman was computed [20], and the socio-economic burden of UI was evaluated in a meta-analysis in 2004 with a cumulative annual expenditure of about EUR 7 billion for Canada, Germany, Italy, Spain, Sweden, and the UK each, and EUR 66 billion for the USA in 2007 [21].

## 3. Stress Urinary Incontinence—Diagnostics and Limitations

### 3.1. Diagnostics of Stress Urinary Incontinence

Correct assessment of the type of incontinence and the extent of incontinence is essential for therapy planning. For all forms of incontinence, a careful anamnesis is the essential basis for the first assessment. In addition to the assessment of risk factors, the effects on different areas of life, including sexuality, should also be examined. In men, attention should also be paid to symptoms of benign prostate syndrome [5,6,7]. Questionnaires to assess the extent and form of incontinence are endorsed [22]. To quantify the exact amount of urine loss, a 24 h pad test is recommended, whereby the patient’s compliance is essential. The same applies to a drink and voiding diary, which should be performed for 48 to 72 h [6].

The following basic diagnoses should be carried out in all patients:Physical examination ○Women: including vaginal examination and stress test (Bonney test)○Men: including prostate examination and stress testUrine analysisUltrasound ○Post-void residual urine volume remaining in the bladder○Men: prostate volume in case of lower urinary tract symptoms (LUTS)
Men: uroflowmetry in case of LUTSQuestionnaire (optional)Drink and voiding diary (optional)


Special diagnostics can be added if it is very likely that they will influence the clinical decision, including:
UrodynamicsCystoscopyPelvic floor ultrasoundQ-Tip test to identify female urethral hypermobilityMarshall–Bonney testDynamic magnetic resonance imaging

Urodynamics are useful if UUI is suspected. Filling cystometry can identify hypersensitivity and overactivity of the bladder muscle, the detrusor. To evaluate the closure mechanism of the urinary bladder, the urethral pressure profile at rest and under stress was described [23]. This is considered a limitation of urodynamics, since the significance of urethral pressure profile measurements were critically assessed [24].

### 3.2. Limitations of Diagnostics of Stress Urinary Incontinence

The limitations in the diagnosis of SUI lie mainly in the fact that the pathophysiology has not been conclusively clarified. Looking at the current literature, a distinction is made between intrinsic sphincter insufficiency and hypermobility of the urethra, particularly in women [2]. In the case of intrinsic sphincter insufficiency, a reduction in muscle cells or their strength is assumed, whereas in the case of hypermobility, the fixation to the symphysis is impaired. Figure 1 gives an overview of urethral hypermobility [25].

Colposuspension and other methods of minimally invasive therapy for women are aimed particularly at the correction of a hypermobile urethra [26,27,28]. During the Burch colposuspension, the bladder neck and the proximal part of the urethra are elevated behind the pubic symphysis to improve the pressure transmission to the urethra. The differentiation between muscular weakness and hypermobility has also gained importance in men. The use of retrourethral functional slings addresses the hypermobile sphincter in particular [29,30].

The difficulty now lies in clarifying the pathophysiology in the context of diagnostics. In men, the perineal elevation test is valued in post-prostatectomy incontinence. Here, external pressure is applied to the perineum under endoscopic vision to visualize the effect of a surgical elevation of the bulbar urethra on the external sphincter. For women, the pelvic floor ultrasound and the Q-Tip test are available. During pelvic floor ultrasound, the mobility of the urethra can be measured in relation to the pubic symphysis and should not be above 2 cm during a Valsalva manoeuvre. The Q-Tip test is performed through the insertion of a Q-Tip in the female urethra; subsequently, the change in the angle of the Q-Tip during a Valsalva manoeuvre is measured. Again, surveying the sphincter complex in such patients by urodynamics is problematic [31], and clinical evidence alone seems to be sufficient to diagnose uncomplicated SUI [32].

## 4. Strategies to Prevent Stress Urinary Incontinence

Considering all diagnostic and therapeutic limitations, the prevention of urinary incontinence is particularly important. This should by no means be a plea for caesarean delivery. Pregnancy by itself bears a corresponding risk due to the continuous load on the pelvic floor [33]. During surgical interventions in the pelvis, the anatomy and function of nerves and muscles should be considered [2,34]. Figure 2 shows the structure of the external urethral sphincter, consisting of striated and smooth muscular components [35]. The horseshoe-shaped sphincter with a pronounced ventral component, in men spanning the prostate and in women lying distally, coincides with the publications of other working groups.

Based on these data, appropriate surgical techniques were adapted for maximum sphincter protection. In addition, innervation should also be considered in terms of SUI prevention [36], and nerve-sparing surgeries that enable continence after interventions [37,38].

## 5. Non-Surgical Treatment of Stress Urinary Incontinence

The non-interventional therapy options for SUI include behavioural therapy measures, pelvic floor training, biofeedback, electrotherapy, and drug therapy approaches [5,6,7]. Men have the option of penis clamps, although their acceptance has fallen significantly [39]. The behavioural therapy options are more limited in SUI than in UUI. The essential aspect here is weight loss [40].

### 5.1. Pelvic Floor Muscle Training

Pelvic floor training aims to strengthen and also target coordination of the muscles of the sphincter complex [41]. It is particularly important in the context of regression after delivery as well as in the context of rehabilitation after radical prostatectomy. An update of the Cochrane database review on the subject of pelvic floor training in women demonstrated its effectiveness, as well as its favourable cost aspects [42]. However, recent reviews on post-prostatectomy incontinence in men failed to come to a clear statement [39,43]. Nevertheless, in clinical practice, pelvic floor exercises after prostatectomy are essential for adequate rehabilitation.

### 5.2. Use of Pessary

A pessary is a device that is inserted into the vagina to treat pelvic organ prolapse or SUI. In pelvic organ prolapse it has shown comparable efficacy in the treatment to pelvic floor muscle training in a randomized controlled trial [44]. While both treatments showed an improvement, surgery was significantly more effective in a recent prospective cohort trial on pelvic organ prolapse [45]. A pessary may also be used in the absence of a prolapse to treat SUI in cases of urethral hypermobility [46]. The device elevates the urethra and results in an increased urethral length through a similar mechanism to colposuspension or slings.

### 5.3. Electrical Stimulation of the Pelvic Floor and Biofeedback

Both electrotherapy and biofeedback are often used in routine clinical practice to exhaust non-interventional measures. In contrast to pelvic floor exercises in women alone, the evidence here must be viewed much more critically. A meta-analysis reported an advantage for electrotherapy as opposed to no or SHAM treatment [47]. But a large, randomized study on pelvic floor training versus pelvic floor training with biofeedback yielded no difference in outcome between the two groups [48].

### 5.4. Medicinal Treatment of Urinary Incontinence

Drug therapy for SUI is clearly limited. In contrast to UUI, with the options of anticholinergics (multiple substances), ß-3 agonists (mirabegron), and the use of a botulinus toxin derivate, onabotulinumtoxin, injected into the detrusor muscle, only one substance is clinically applied to treat SUI nowadays [6]. Duloxetine has been approved for female SUI based on phase III studies comparing duloxetine to a placebo with an improvement in urinary incontinence and quality of life [49,50], and an additional positive effect of pelvic floor training in combination with duloxetine [51]. Although studies show the effectiveness in patients with post-prostatectomy incontinence [52], duloxetine is still not approved for SUI in men. 

Due to the health risks, there are clear limits to the use of duloxetine to treat SUI. The same side effects exist as with other selective serotonin reuptake inhibitors in the therapy of depression, which can also be therapy-limiting in the case of marginal urinary incontinence [52]. In addition, the patient must be given detailed information that the therapy is purely symptomatic (stimulation of the innervation from the Onuf’s nucleus) and that no cure can be achieved here. Therefore, a permanent therapy is necessary; this aspect should be taken into account when deciding on a form of therapy [2,6].

#### 5.4.1. Treatment of Urge Urinary Incontinence by Anti-Cholinergic Drugs

Muscarinic receptors play an important role in signal transduction in the body. Muscarinic receptors are integrated in G-protein-coupled complexes in cell membranes, and relay acetylcholine signals into cells by secondary messengers such as cAMP, diacylglycerol (DAG), inositol-1,4,5-trisphosphate (Ins3P or IP3), or others. In the detrusor muscle, the main subtypes are M_2_ and M_3_. The main function of M_2_ receptors in the bladder is micturition. The binding of acetylcholine to the receptor triggers the contraction of the detrusor to enable voiding. An overactivity of the detrusor, irrespective of its origin, can result in the sensation of urgency and UUI. Selective anticholinergic drugs that can act as competitive antagonists at the receptor have been established as medical treatments.

During the storage phase, the stretching of the urothelium will induce a release of acetylcholine. This might be responsible for spontaneous contraction in overactive bladders. The relatively low amount of acetylcholine compared to the release through parasympathetic nerves can be inhibited by a competitive antagonist. In comparison, the release of acetylcholine during voiding contraction is so high that anti-muscarinergic drugs in the therapeutic dose range have no effect on micturition.

#### 5.4.2. Treatment of Urge Urinary Incontinence by ß-3 Agonists

Storage of urine in the bladder is facilitated by the sympathetic nervous system. Adrenalin or noradrenalin binds to ß-adrenoreceptors expressed on different cells, including smooth muscle cells. Smooth muscle tissue is part of the urethral closure complex and is also found in the detrusor muscle. Therefore, ß-3 agonists constitute an alternative in the medical treatment of urge urinary incontinence. In the human bladder, the ß3-adrenoreceptor subtype is mainly responsible for effectuate relaxation of the detrusor muscle. The active substance mirabegron acts as a selective agonist to the ß-3 receptor. Through this, mirabegron achieves increased bladder capacity. The ß-3-adrenoceptor is expressed in both the detrusor muscle and the urothelium. The function of the receptors in the epithelium is not yet known, but the intravesical instillation of ß3-adrenoceptor agonists decreased the voiding frequency in rats and had an additive effect on intravenous application [53]. This suggests that, comparable to muscarinic antagonists, ß3-agonists achieve effects both in the afferent and efferent limb of the voiding process.

#### 5.4.3. Treatment of Urge Urinary Incontinence by Injections of Botulinum Toxin A

If anticholinergic drugs and ß-3 adrenoceptor agonists are ineffective, injection of botulinum toxin A is a third option in current clinical practice. The neurotoxin is produced by different bacteria, primarily *Clostridium difficile*, and especially the suborder A is used in human medicine. Botulinum toxin inhibits the release of acetylcholine in the presynaptic neuron and, thus, generates paralysis in the affected muscle.

The best-studied effect is the cleaving of SNARE proteins, which are essential for the binding of the synaptic vesicles containing acetylcholine to the plasma membrane. Recent studies have shown that repeated botulinum toxin injection into the detrusor muscle not only reduces the expression of SNAP-25, as part of the SNARE complex, but also reduces inflammation. Decreased tryptase activity, fewer apoptotic cells, and the apoptotic signaling molecules Bax and p-p38 were observed after three repeated botulinum toxin injections [54].

#### 5.4.4. Future Pharmacological Regimen in Research or under Preclinical Investigation

Most current studies focus on new ways to deliver active ingredients to the urothelium and detrusor muscle, and some new forms of treatments are being investigated. A liposomal formulation of botulinum toxin A as an instillation therapy has shown a clinical effect on an overactive bladder, but no effect was seen on the expression of SNAP-25. This suggested that the effect on the paralysis of the detrusor is limited, and the results might be caused by reduced afferent signals from the urothelium [55].

Bethanechol is a direct-acting parasympathomimetic medication that stimulates the detrusor muscle. So far, it is not recommended by the international continence society guidelines in non-obstructive urinary retention or underactive bladder, but might help in decatheterization tests. Further tests in the dosage of 50mg/BID are warranted [56].

An entirely new approach is URO-902, a plasmid vector expressing the human large-conductance Ca^2+^-activated K^+^ channel. Activation of this channel has been shown to reduce smooth-muscle cell excitability. A phase I study promises first efficacy and safety data through either instillation or injection of URO-902; possible therapy costs due to this medication will have to be evaluated [57].

#### 5.4.5. Novel Approaches to Treat Stress Urinary Incontinence

Recently, a web-based study investigated molecular targets to treat lower urinary tract (LUT) dysfunction in the future [58]. Three main targets—the bladder, urethra, and nerves—were explored. The status of possible future drugs or targets was included: proof of concept studies versus promising data based on preclinical studies [58]. It is not surprising that for future LUT therapies, several molecular targets, including receptors for adenosine A1, neurotensin, bombesin, and cannabinoids were discussed. They may act on the bladder wall, and/or urethra, and via modulating the nervous regulation of voiding. However, as stated above, the etiology of SUI is associated with mechanical stress, tissue injury, and aging. SUI, therefore, develops at least in part due to insufficient wound healing after an injury or impact and it progresses in time during aging. Structural changes such as loss of muscle tissue and fibrosis are among the consequences and, thus, reduce contraction and control of the urethral sphincter. Therefore, for a long time and until recently, treatment of SUI with low molecular weight drugs including hormones was not even considered a treatment option by experts in the field [4,6]. Moreover, earlier trials along these lines were not encouraging. For instance: the chance of suffering from an SUI was raised in multiparous women in menopause. It is known that the expression of elastin and other components of the extracellular matrix changes in the elderly. In women, this change is associated with hormonal changes in menopause. Therefore, administration of, for instance, estrogen was investigated to treat SUI or at least to ameliorate SUI symptoms. Although the women treated reported subjective improvement, objective urological parameters (e.g., urinary diary, pad test, quality of live reports by patients, etc.) did not reach significance [15,59,60]. These studies were, therefore, not continued.

In contrast, recent studies investigated the molecular pathways involved in SUI therapy employing electrophysiological stimulation of the lower pelvic floor muscles. In sports, electrophysiological stimulations are well known for their efficacy in relaxing muscles after exercise, activation of local blood circulation, and reduction of pain symptoms, thus supporting the athletes’ regeneration. This knowledge was now relayed back to pre-clinical animal studies of SUI therapy. In mice, SUI-like symptoms were induced by vaginal distension. Then, electrophysiological stimulation was performed for a follow-up period of up to seven days [61]. The effects of vaginal distension and its regeneration by electrophysiological stimulation in this SUI model were monitored and standard measures employed in the clinical routine were applied. Accordingly, the maximum bladder capacity and the leak point pressure were determined. The voiding upon sneezing served as a surrogate for a pad test [61]. The electrophysiological stimulation of incontinent mice improved the maximal bladder capacity, leak point pressure, and urine loss upon sneezing significantly. This animal model allowed, in addition, histological analyses of the tissue treated. Here—depending on the electrophysiological stimulation regimen—a robust to significant induction f expression of type I collagen was recorded. The expression of transforming growth factor beta-1 (TGF-β1) and the activation of Sma-and-mothers-against-decaplegic-2 and -3 (SMAD-2; SMAD-3) were significantly reduced after vaginal distension. However, the phosphorylation of the SMADs increased significantly after electrophysiological stimulation [61]. The reduction of activated phospho-SMAD-2 (p-SMAD2) and p-SMAD-3 is not unexpected, as these intracellular second messengers are specific intracellular modules involved in signal transduction of the TGF-ß pathway (Figure 3). Moreover, TGF-β1 is a key regulator affecting cell differentiation and proliferation and promotes expression of components of the extracellular matrix [62].

Along these lines, TGFs have been shown to stimulate smooth muscle differentiation and chondrogenesis from mesenchymal progenitor cells, at least in vitro [63,64]. These processes take, under optimal conditions in vitro, at least 3–4 weeks of induction. Moreover, there is excellent experimental evidence that mesenchymal stromal cells, also referred to as mesenchymal stem cells, do not undergo efficient differentiation along the chondrogenic or myogenic or in other lineages in vivo, with the exception of osteogenesis [65]. Differentiation of cells is therefore unlikely to contribute to the amelioration of SUI-like symptoms described above [61]. But other mechanisms come, here, into play: TGF is well described as a cytokine promoting the expression of type I collagen [66]. The enhanced amounts of TGF after electrophysiological stimulation may, therefore, enhance the expression of collagen. But it is not clear how electrophysiologic stimulation caused the increase in TGF available locally in the urethra. Several mechanisms may contribute here: electrophysiological stimulation causes enhanced circulation and, thereby, may make TGF available locally. Wound healing processes may cause tissue degradation, thus setting matrix-bound TGFs free. This mechanism was described in bone regeneration [67] and may contribute to the TGF-mediated SMAD phosphorylation in this SUI model as well. But more studies are required to fully understand the mechanism of enhanced wound healing after electrophysiological stimulation on a molecular level.

Recently, a study corroborated that electrophysiological stimulation of human vaginal-wall-derived fibroblasts enhanced the mRNA transcription of TGF- β1 and type I and type III collagens [68]. Interestingly, in fibroblasts derived from SUI patients, expressions of integrin β1 (alias CD29), TGF- β1, and type I collagen were significantly lower when compared to fibroblasts derived from donors not suffering from SUI [68]. This indicated that the changes observed in the expression and regulation of collagen and TGF in the urethra were not locally restricted but were detectable in fibroblasts from neighboring sites as well. Pretreating these fibroblasts with an antibody to integrin β1 was associated with a reduced response of the cells to the electrophysiological stimulation. This indicated that the integrity of cell–matrix interactions may contribute to the sensitivity towards electrophysiological stimulation. Intracellular integrin signaling may play a role in this context (Figure 4). Integrins are heterodimeric receptors consisting of 1 of 18 alpha and 1 of 8 beta chains. Integrin β1 is part of many such heterodimers binding, for instance, to a peptide consisting of arginine–glycine–aspartic acid (RGD). The RGD peptide provides a positive charge, a kink, and a negative charge, ideal for strong electrostatic interactions of the peptide with its corresponding receptors. This RGD motif is found in many proteins of the extracellular matrix, including collagen, fibronectin, vitronectin, bone sialoprotein, and others. Using other peptide motives, the integrin β1 is also part of receptors for fibrinogen, tenascin, and VCAM-1. Thus, reacting integrin β1 with a specific antibody interferes with structural changes, attachment, and migration of cells, and with gene expression involved in the regulation of cell survival and gene expression (Figure 4). The molecular link between the reduced sensitivity of cells toward electrophysiological stimulation after incubation with an anti-integrin antibody remains to be explored in future studies.

Activation of blood circulation may also contribute to another regimen of experimental SUI treatment. In adipose versus lean rats, urinary leak point pressure was compared, and fat rats had a reduced leak point pressure [69]. Treatment of the lower body of adipose rats with mechanical pulses significantly increased the leak point pressure, although significant changes in the muscular thickness of the urethral sphincter complex were not observed by immunohistochemistry in lean rats. In adipose rats, the gain of muscle tissue did not reach significance. But the number of muscular progenitor cells increased in the urethra significantly [69]. We hypothesize that mechanical impulses may have activated resident satellite cells to initiate proliferation and differentiation. This aspect may be important for ongoing clinical studies using a combined therapy of myoblast injections and electrophysiological stimulation (see Schmid et al. in this Special Issue).

Yet another preclinical study investigated the role of myostatin in sphincter regeneration. Myostatin is a soluble member of the TFG-β family of proteins. It is a 43 kDa protein that forms a homodimer that binds to receptors of the TGF receptor family. Myostatin activates SMAD signaling and kinase pathways (Figure 5). In serum or outside the cell, it may also interact with soluble inhibitors that then prevent binding to the ActRIIB receptor on cell surfaces (not shown). Myostatin negatively regulates myoblast proliferation and differentiation to myofibers. Mutations affecting its binding to the receptors are therefore associated with muscle hypertrophy [70]. Myostatin modulates the expression of MyoD, myogenin, and Myf5, key factors required for the differentiation of skeletal muscle cells from myogenic progenitors, also referred to as satellite cells (Figure 6). In a recent study, obese rats were subjected to a functional knock-out of myostatin, thus enhancing the expression of myogenic differentiation factors [71]. As shown above, obese rats presented with reduced leak point pressure, corroborating other studies [69]. After silencing the expression of myostatin, enhanced myogenesis was observed, and the leak point pressure, urethral continence, and thickness of the striated urethral muscle, as well as the ratio of smooth versus striated sphincter muscles, were significantly improved [71]. Tissue engineering with myostatin silenced or knock-out cells is regarded as a technically very demanding regimen and, therefore, will possibly not be accepted for entry in clinical feasibility studies. However, the animal experiments with obese rats may pave the way for future medical treatment aiming at transient and local reduction of myostatin to enhance the proliferation and differentiation of urethra resident satellite cells [72].

## 6. Surgical Treatment of Stress Urinary Incontinence

### 6.1. Interventional Treatment of Female Stress Urinary Incontinence

Interventional procedures for the treatment of SUI in female patients include different surgical options [2,6,8]. But a prolapse of an SUI patient should be treated in advance or in an experienced center, contemporaneously [73]. The potential of laser therapy cannot be conclusively assessed at present. This technique needs further studies [74].

Figure 7 gives (except for bulking agents) an overview of the various operational techniques that are presented below. It should be noted that tension-free tapes and colposuspension address urethral hypermobility, whereas bulking agents, autologous fascia slings, and artificial sphincter muscles focus on intrinsic sphincter insufficiency.

#### 6.1.1. Use of Bulking Agents in Female Stress Urinary Incontinence

The idea of bulking agents is to coaptate the insufficient sphincter muscle with a “collar” so that a corresponding active muscular effect can be achieved again. The long-term benefits of bulking agents seem limited [76]. The main advantage is the low level of invasiveness [6]. A retrospective study with a seven-year follow-up showed relief or reduction of incontinence in 67% of patients, in 11.1% there was no change, and 2.3% worsened. 19.5% needed additional surgical treatment for persistent SUI [77].

#### 6.1.2. Minimally Invasive Slings to Treat Stress Urinary Incontinence in Females

For several decades, transvaginal tension-free tapes (TVT) have become an established therapy for female UI. Comparable results between colposuspension and TVT implantation were observed [78]. Several products for retropubic or transobturatoric application are available, each with its advantages and disadvantages [79,80]. But transvaginal access yields side effects affecting sexuality, and the implantation of non-absorbable slings or meshes for descensus may cause complications by material stiffness or corrosion [81]. Therefore, the US Food and Drug Administration issued a warning. This led to a change in surgical strategies for women and yielded a revival of classic surgical techniques such as colposuspension [9]. The use of tapes for incontinence treatment is regarded as a sensible and legitimate alternative, provided that patients are informed accordingly [82,83].

#### 6.1.3. Open and Laparoscopic Intrapelvic Surgery to Treat Stress Urinary Incontinence in Females

Originally, colposuspension was the standard treatment for SUI in women. The Burch colposuspension is one of the most common procedures: non-resorbable sutures lift the lateral vaginal ligament to the upper pubic bone fascia like a hammock. Publications underline the effectiveness of open colposuspension, but also emphasize the lack of long-term data on laparoscopic operations [84]. Other studies reported no differences in terms of effectiveness, patient satisfaction, or complications, although patients with colposuspension had an increased prolapse risk [85].

#### 6.1.4. Artificial Urinary Sphincter in Female Patients with Stress Urinary Incontinence

The artificial urinary sphincter (AUS) has been the established procedure for treating SUI in men since 1973. In women, AUS implantation is carried out on the bladder neck via a completely retropubic approach. The pump is inserted into one of the labia majora for operation. Artificial urinary sphincter implantations vary in frequency between countries. A small study achieved success in more than 84% of patients treated at a follow-up of 18 months [86]. In summary, the artificial sphincter is currently considered a salvage option for the treatment of SUI in women. Possibly, complication rates could be reduced if an earlier application takes place, with which fewer adhesions in the vesicovaginal space are to be expected.

### 6.2. Surgical Treatment of Male Stress Urinary Incontinence

Surgical therapy options are only used when conservative options are exhausted. In particular, after radical prostatectomy, incontinence operations including bulking agents, band systems, periurethral balance, and artificial sphincters should be indicated after 12 months at the earliest [30]. For many years, the artificial sphincter has been the gold standard of therapy for stress urinary incontinence (see below) [87].

#### 6.2.1. Use of Bulking Agents in Male Stress Urinary Incontinence

Injections of bulking agents were often used in men after post-prostatectomy incontinence [76]. Our own sobering experiences coincide with the literature: the outcome of bulking agents in patients with incontinence after radical prostatectomy is poor [88]. Although the procedure impresses with low invasiveness, the long-term successes, in particular, seem low. In addition, injections close to the sphincter have potentially bad effects on urethral mobility.

#### 6.2.2. Minimally Invasive Slings to Treat Stress Urinary Incontinence in Males

Minimally invasive slings can be offered in adjustable and non-adjustable forms for patients with mild to moderate post-prostatectomy incontinence, a sufficiently mobile urethra, and a morphologically intact sphincter [29]. Figure 8 illustrates such an implant and the retrourethral positioning with transobturatoric routing. Cure or improvement was reported in 76.9% of patients 12 months and in 76.8% of men three years after treatment. The complication rate was low [89]. A history of radiation therapy to the prostatic bed has a negative impact on success.

In contrast to the functional non-adjustable sling, all adjustable band systems are placed suburethrally. They have an obstructive effect but no influence on urethral mobility. Currently, there are no randomized controlled trials published comparing the different slings. An example of such a sling is presented in Figure 9. In a meta-analysis of data from 1919 patients from 29 studies, some slings appeared to be more effective and resulted in fewer explanations when treating post-prostatectomy incontinence [90]. Other studies compared other implants and reported on low incontinence and explanation rates as well [91]. In summary, adjustable ligaments offer patients the option of adaptation in the longer term; however, compared to artificial sphincters, the data are sparse, and comparative studies are pending.

#### 6.2.3. Artificial Urinary Sphincter in Male Patients with Stress Urinary Incontinence

The first description of the artificial urinary sphincter (AUS) was presented in 1973 [92]. Since then, a variety of implants have been developed. Some became the gold standard for treating male SUI for many years. Figure 10 shows an example of such an implant and the positioning of its components.

Sphincter implants are available either with or without antibiotic coating, although a significant reduction in infections by such coating was not observed [93]. But data suggest that the surgeon’s experience has a significant influence on the outcome, as complications were significantly lower in high-volume centers [94]. AUS reduces any degree of incontinence with a success rate of 80% with grade 2 and grade 3 incontinence. Prior irradiation is not a general contraindication. The main disadvantage is the high revision rate, which is sometimes over 20% [6]. In addition to erosions requiring explanations, revisions often necessitate further complex surgeries to gain at least some sort of continence. Novel implants address this aspect with an improved design [95]. Figure 11 demonstrates such a novel implant and its genitourinary positioning. Current studies show promising results, but long-term data are still missing [96,97].

## 7. Summary of Limitations of Current Stress Urinary Incontinence Treatment

Little has changed in the last decade with regard to the basic therapy options [8]. One of the main focuses of research was on the development of adjustable incontinence implants in view of the knowledge that the reduced success rates would otherwise make surgical interventions necessary. Nevertheless, the published case numbers for individual implants, especially in men, are too low to be able to make clear statements about long-term success and the rate of complications. The lack of randomized controlled studies, which are essential for the assessment of a form of therapy and for adequate approval according to current standards of the Medical Devices Act, is particularly critical [98]. The decision of the FDA to provide tapes and meshes with a very clear safety warning was certainly far-reaching. In the area of incontinence, this mainly affected the colposuspension techniques.

None of the therapy options listed above—with the exception of pelvic floor training—has the potential to causally cure stress urinary incontinence with a high success rate. The last two decades have shown that approaches in regenerative medicine or cell therapy cannot easily be established in clinical routine, despite many preclinical studies and efforts, partly due to a lack of success and partly due to a lack of compliance with regulatory aspects. But more recently, promising feasibility studies with small cohorts were conducted [3,99,100]. Thus, cell therapies may become an option in the future.

Finally, it should not go unmentioned that numerous approaches to therapy for stress urinary incontinence focus either on intrinsic sphincter insufficiency or on hypermobility of the urethra. Aspects of the innervation of the sphincter muscle that are partly responsible for incontinence in some of the patients (neurogenic underlying disease, stretching of the pudendal nerve during childbirth…) are left completely out of focus. At best, drug therapy should be mentioned here, which, by strengthening the neurogenic component, brings improvement for some of the patients [101].

## 8. Conclusions

Current therapy options for stress urinary incontinence in men and women can improve continence and, thus, optimize quality of life. Often, this goal is only possible with the use of implants, which, in turn, present risks perioperatively, but also in the long term. However, the fundamental problem does not concern the forms of therapy alone; rather, diagnostics are often not able to clearly work out the pathophysiology of incontinence in the individual patient. The wishes for the future are therefore both the further development of diagnostic possibilities and the focus on a causal curative therapy approach—be it with conventional possibilities or the promising options of stem cell therapy or tissue engineering [102,103].

## Figures and Tables

**Figure 1 biomedicines-11-02486-f001:**
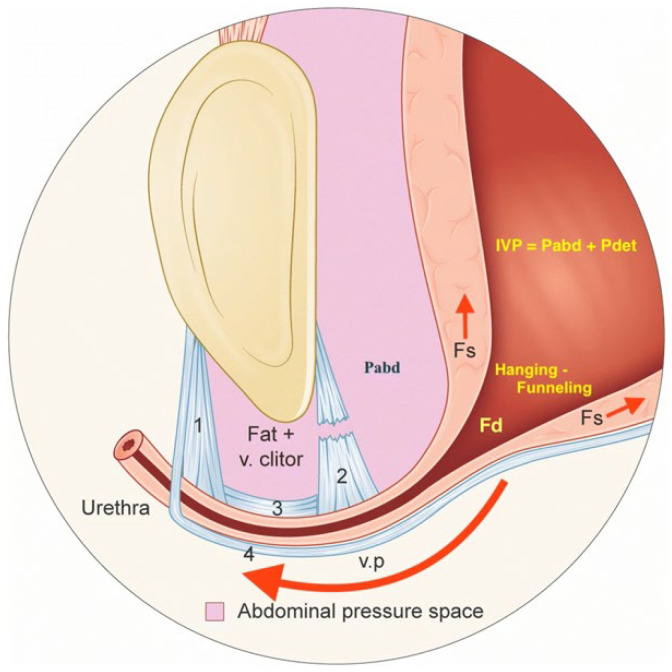
Hypermobility of the proximal female urethra caused by the release of the attachments to the symphysis pubis/pubic bone. Published by Bergstrom under Creative Commons CC BY license [25]. IVP: intravesical pressure; Pabd: intrabdominal pressure; Pdet: detrusor pressure; Fs: shearing force; Fd: outflow distending force; v. clitor: vena clitoridis; v.p: vaginal point 1: right anterior pubourethral ligament; 2: right posterior pubourethral ligament; 3: right intermediate pubourethral ligament; 4: pubocervical fascia.

**Figure 2 biomedicines-11-02486-f002:**
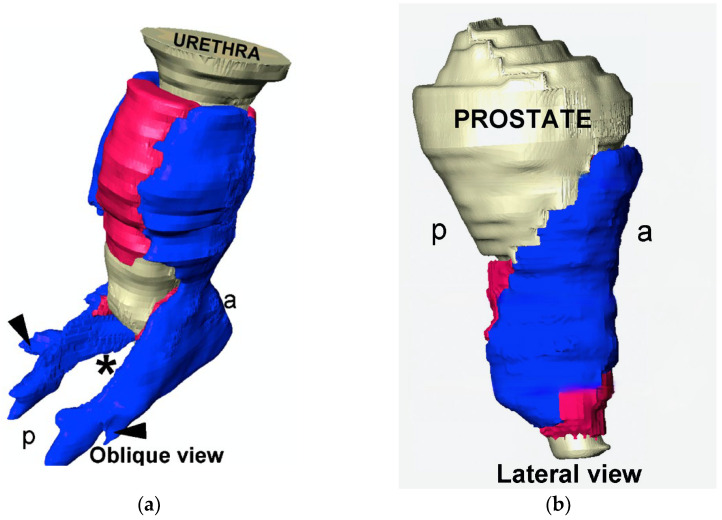
External urethral sphincter consisting of external striated in blue (rhabdosphincter) and internal smooth muscle in pink (lissosphincter); prostate and urethra in grey ((**a**) male, (**b**) female); p: posterior; a: anterior. Permission granted by Elsevier [35].

**Figure 3 biomedicines-11-02486-f003:**
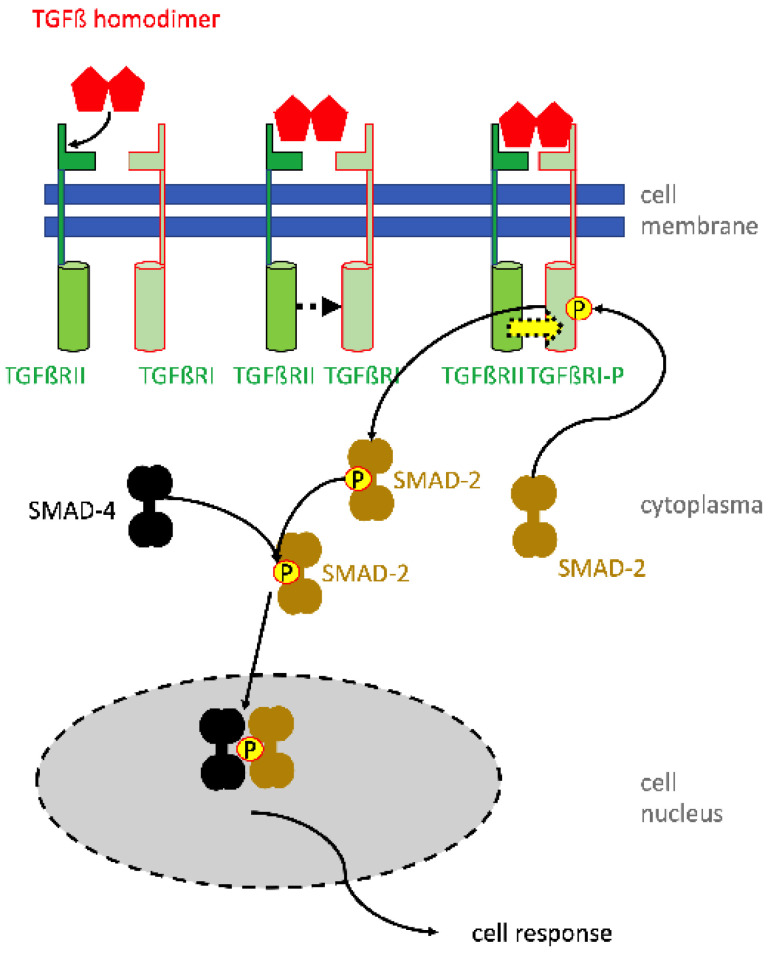
Activation of cells by TGF-beta using the SMAD signaling pathway. The homodimer of TGF-beta binds first with low affinity to the TGF-beta receptor 2 (TGFbRII) (top left). This causes a conformational change facilitating an approximation of the TGFßRII molecule to TGFßRI (top middle). Adjoining TGFßRs form a high-affinity receptor binding TGFß at the binding epitopes of both receptor components (top right). The close heterodimeric receptor enables the intracellular kinase domain of TGFRII to phosphorylate and thereby activate the corresponding domain of TGFßRI (yellow arrow). Activated TGFßRI interacts with SMAD-2 (or SMAD-3) and phosphorylates SMAD-2 or -3 (below). Phosphorylated SMAD-2 then interacts with SMAD-4. The pSMAD-2/SMAD-4 complex translocates in the cell nucleus and there interacts with transcription initiation and elongation factors to activate the gene expression of the respective target genes. The TGF-mediated intracellular signaling via MAP-kinases (e.g., ERK, p38) and Pi3 kinase, regulating downstream AKT and TOR, are omitted in the graph, but play an important role in the anti-apoptotic action of TGF.

**Figure 4 biomedicines-11-02486-f004:**
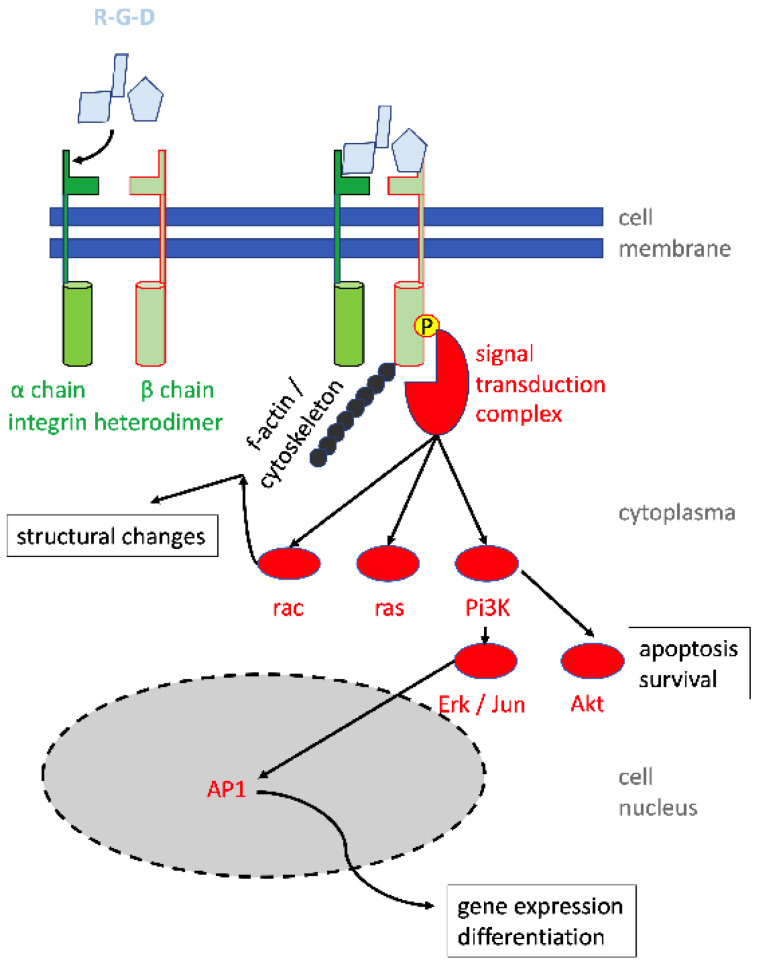
Integrin signaling and regulation of structural changes, cell survival, differentiation, and gene expression. Peptides (light blue) bind first to the integrin α chain, causing a conformational change, thus facilitating an approximation to the integrin β chain. A high-affinity integrin complex is generated (green), and both cytoskeletal components (e.g., f-actin; black) as well signal transduction proteins (e.g., vinculin, paxillin, talin, focal adhesion kinase; red) bind to the intracellular domain of the integrin β chain. Secondary intracellular signal pathways include rac, ras, pi3K, erk, akt, and others. They relay signals, for instance to AP1 binding sites in integrin-regulated promotor sequences.

**Figure 5 biomedicines-11-02486-f005:**
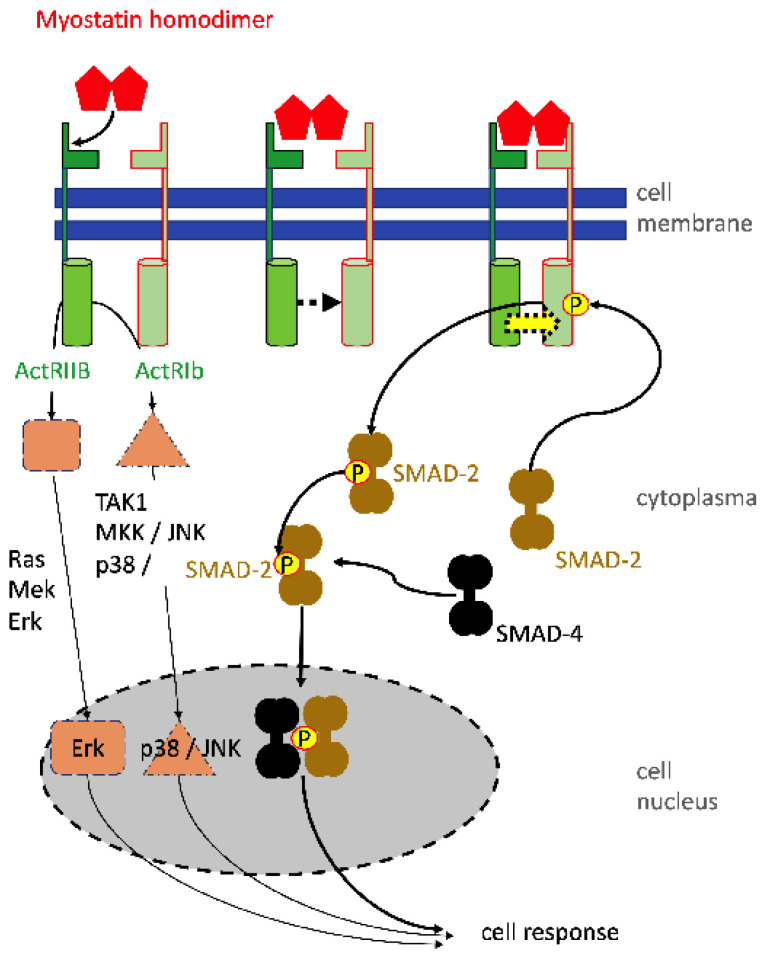
Myostatin-dependent cell signaling involves SMAD and mitogen-activated protein (MAP) kinase pathways. Myostatin id, a member of the TGF-beta family, therefore uses basically the same signal transduction pathways (compare Figure 1). A significant difference to other members of the TGF family of cytokines is that myostatin inhibits proliferation and differentiation of myogenic progenitor cells. In addition to the classical SMAD-dependent signaling, myostatin binding to the Act receptor II B induces activation of two kinase pathways relaying signals via Ras–Mek-Erk1/2 and Tak1–Mkk to p38MAP kinases or through Jnk. Erk, p38MAP kinase, and Jnk interact in the cell nucleus with transcription factors and there modulate the expression of myostatin target genes.

**Figure 6 biomedicines-11-02486-f006:**
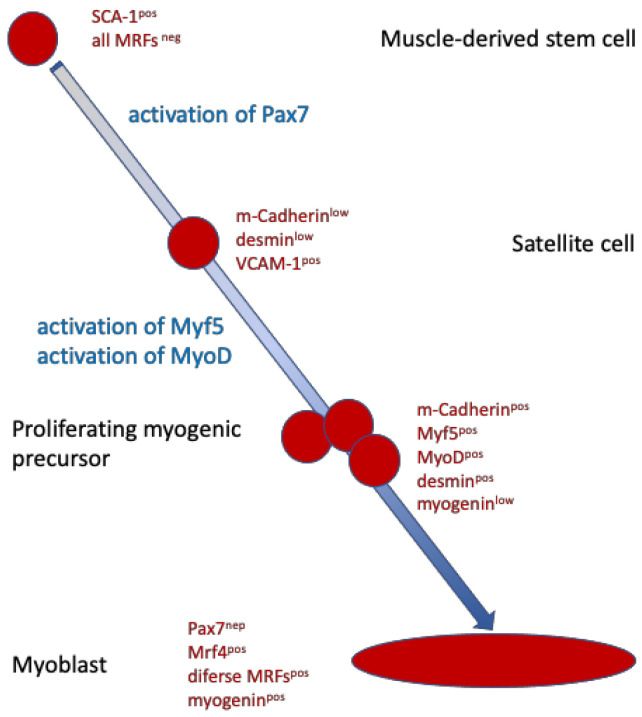
Simplified scheme of muscle cell differentiation. Muscle-derived stem cells are quiescent cells but already pre-programmed towards myogenic differentiation. Upon activation of transcription factor Pax7, muscle stem cells become satellite cells. Satellite cells reside in skeletal muscles. Activation by hypoxia, stress (muscle injury), or by certain cytokines activates satellite cells to then express myogenic differentiation factors such as MyoD. After continued stimulation by cytokines and tissue regenerative signals, proliferation of myogenic precursor cells is elevated, and such cells form myoblasts that fuse to multinuclear myofibers.

**Figure 7 biomedicines-11-02486-f007:**
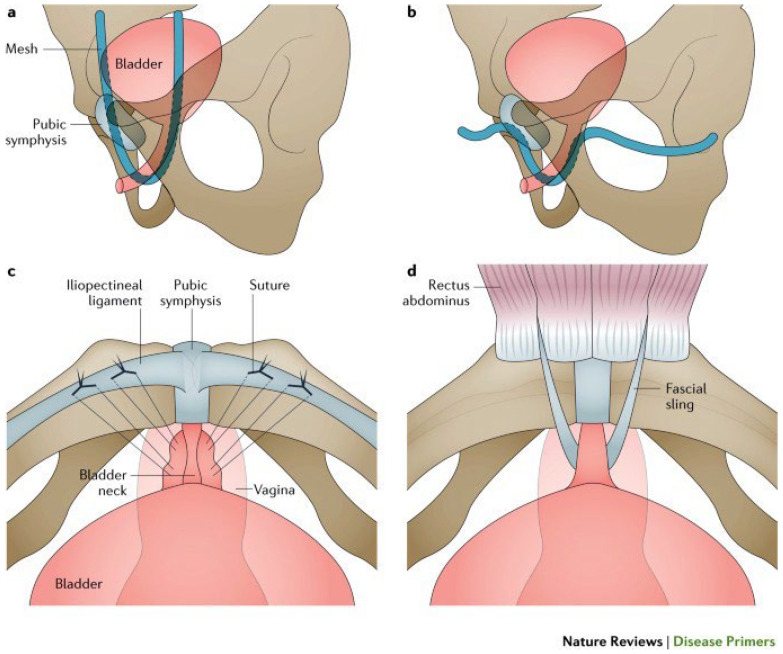
Illustration of different surgical techniques to treat female stress urinary incontinence. (**a**) TVT, (**b**) TVT-O, (**c**) Burch colposuspension, and (**d**) autologous fascial sling. Permission granted by Springer Nature [75].

**Figure 8 biomedicines-11-02486-f008:**
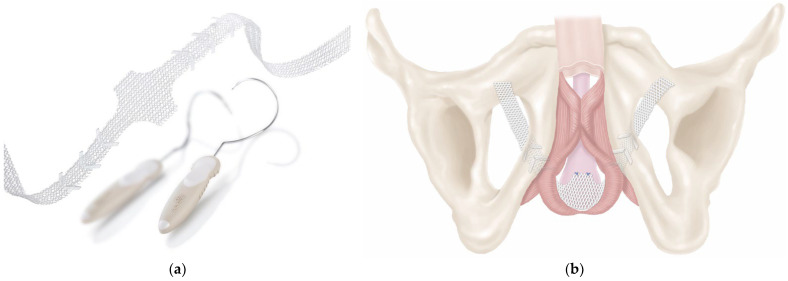
Retrourethral functional sling (Advance XP^®^) to treat mild to moderate post-prostatectomy stress urinary incontinence ((**a**) sling and introducers, (**b**) illustration of sling placement). Permission granted by Boston Scientific Medizintechnik GmbH, Ratingen, Germany.

**Figure 9 biomedicines-11-02486-f009:**
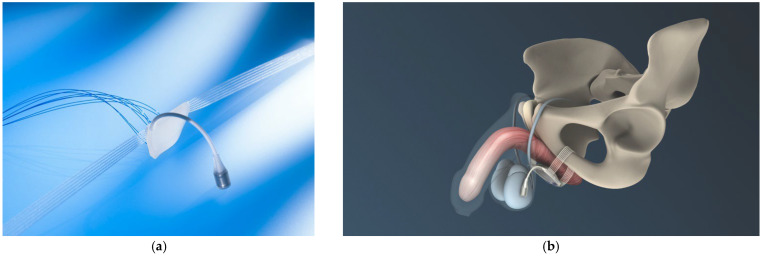
Suburethral adjustable sling (ATOMS^®^) ((**a**) sling with water-filled cushion and port, (**b**) illustration of sling placement). Permission granted by A.M.I. GmbH, Feldkirch, Austria.

**Figure 10 biomedicines-11-02486-f010:**
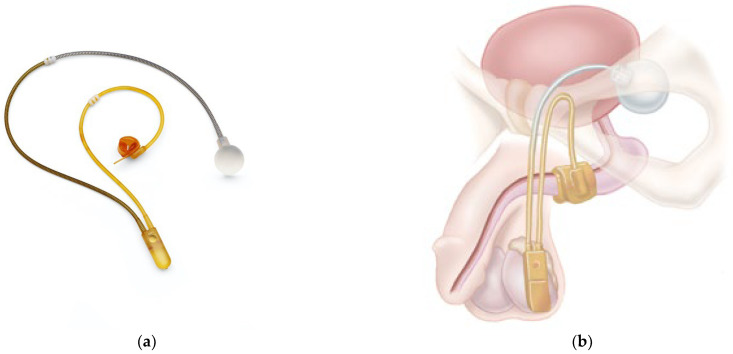
Example of an artificial urinary sphincter ((**a**) three-piece implant, cuff and pump covered with antibiotics, (**b**) illustration of the artificial urinary sphincter in place). (AMS 800^®^, permission by Boston Scientific Medizintechnik GmbH, Ratingen, Germany).

**Figure 11 biomedicines-11-02486-f011:**
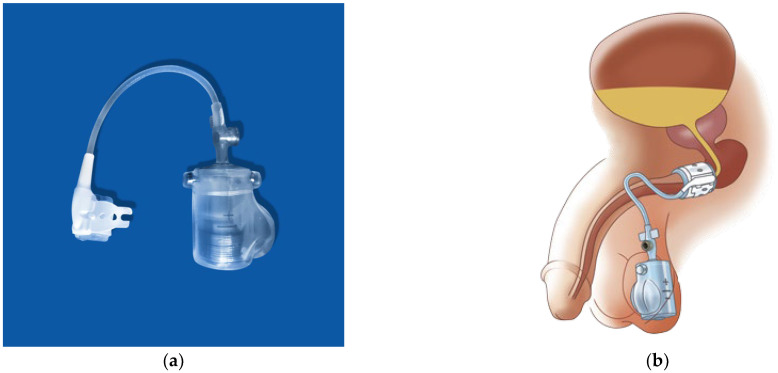
Examples of an adjustable artificial urinary sphincter implant. (**a**) The implant; (**b**) illustration of adjustable artificial urinary sphincter in place (ZSI 375^®^; permission by Zephyr Surgical Implants, Les Acacias Genf, Switzerland).

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
