# Peer review of "Stress Urinary Incontinence: An Unsolved Clinical Challenge"

_biomedicines, 2023, doi:10.3390/biomedicines11092486_

Round 1

Reviewer 1 Report

The authors have conducted a valuable review on the topic of stress urinary incontinence. However, I would like to underscore a few points that I believe require attention.

I would like to emphasize the difficulties I encountered while trying to comprehend the section explaining the pathophysiology of urinary incontinence.

Moreover, I would like to highlight some minor shortcomings that I cannot overlook.

-In Figure 1, while the image is properly positioned and anatomically sufficient, the absence of explanations for the abbreviations within the image has led to difficulties in comprehension.

-Likewise, abbreviations and colors are not clarified in Figure 2.

Author Response

We thank the reviewer for the valuable comments on our manuscript. We addressed the points, changes to the manuscript are marked in yellow in the version submitted.

  1. I would like to emphasize the difficulties I encountered while trying to comprehend the section explaining the pathophysiology of urinary incontinence.

Reply: Thank you for pointing this out. We rephrased some sentences to improve readability.

  1. a) In Figure 1, while the image is properly positioned and anatomically sufficient, the absence of explanations for the abbreviations within the image has led to difficulties in comprehension.

          b) Likewise, abbreviations and colors are not clarified in Figure 2.

Reply: We thank the reviewer for the important comment. We have added the missing labelling and explanation of the abbreviations in Figure 1 and 2.

Reviewer 2 Report

Thank you for this important paper, Stress Urinary Incontinence: An Unsolved Clinical Challenge. This review is well written and illustrated. There are a few suggestions:

1.      For SUI, intrin- 38 sic and extrinsic incontinence were discussed as they offer different options for therapy 39 (4)  - possibly change to are discussed because this is your paper.

2.      They have not directly improved the number and 52 function of the sphincter muscle fibers, nor its vascularisation or enervation. – very interesting, but, I thought I read that function has been shown to be improved – please clarify.

3.      Line 77, could you also comment on percent seeking treatment?

4.      Figure 1, can you add what 1 – represent and other abb?

5.      Line 133 – please define – colposuspension

6.      Line 140, Can you define (clarify) - perineal elevation test is valued in post-prostatectomy incontinence. For 139 women, the pelvic floor ultrasound and the Q-Tip test are available

7.      Fig 2, please add abb and arrow meaning in legend

8.      191 revise . But Duloxetine is still not approved for 191 SUI in men. Although studies show the effectiveness in patients with post-prostatectomy 192 incontinence (48).      TO: Although studies show the effectiveness in patients with post-prostatectomy 192 incontinence (48), Duloxetine is still not approved for 191 SUI in men.

9.      205 - Accordingly, long-term therapy should be sought, and this 205 aspect should be included in the indication (3, 7).   Not clear, please revise

10.   222 – might you consider further study for Bethanechol as recently suggested – 50 mg/BID needs further testing! Walter, J.; Wheeler, J. Use of Bethanechol, 50 mg/BID, for a Failed Decatheterization Test: A Position Statement. Uro 2022, 2, 134–136.   https://doi.org/10.3390/uro2020016

11.    236.  However, very expensive

12.   Please add a section on pessary use and outcomes – this should also be first-line therapy.

good

Author Response

We thank the reviewer for the valuable comments on our manuscript. We addressed the points, changes to the manuscript are marked in yellow in the version submitted.

  1. For SUI, intrin- 38 sic and extrinsic incontinence were discussed as they offer different options for therapy 39 (4)  - possibly change to are discussed because this is your paper.

Reply: We thank the reviewer for his attentive review. We have adjusted the wording.

  1. They have not directly improved the number and 52 function of the sphincter muscle fibers, nor its vascularisation or enervation. – very interesting, but, I thought I read that function has been shown to be improved – please clarify.

Reply: We thank the reviewer for the very good note. We have clarified the effects of the individual treatment methods on the sphincter.

  1. Line 77, could you also comment on percent seeking treatment?

Reply: We thank the reviewer for the valuable comment. We have added a reference to the issue of those affected seldomly seeking treatment.

  1. Figure 1, can you add what 1 – represent and other abb? Fig 2. please add abb and arrow meaning in legend.

Reply: We thank the reviewer for the important comment. We have added the missing labelling and explanation of the abbreviations in Figure 1 and 2.

  1. Line 133 – please define – colposuspension

Reply: We thank the reviewer for the valued comment. We have added a short section explaining the colposuspension.

  1. Line 140, Can you define (clarify) - perineal elevation test is valued in post-prostatectomy incontinence. For 139 women, the pelvic floor ultrasound and the Q-Tip test are available

Reply: We thank the reviewer for the valued comment. We have added a short section explaining the perineal elevation test, pelvic floor ultrasound and Q-Tip test.

  1. 191 revise . But Duloxetine is still not approved for 191 SUI in men. Although studies show the effectiveness in patients with post-prostatectomy 192 incontinence (48).      TO: Although studies show the effectiveness in patients with post-prostatectomy 192 incontinence (48), Duloxetine is still not approved for 191 SUI in men.

Reply: We thank the reviewer for his attentive review. We have adjusted the wording.

  1. 205 - Accordingly, long-term therapy should be sought, and this 205 aspect should be included in the indication (3, 7).   Not clear, please revise

Reply: We thank the reviewer for his attentive review. We have adjusted the wording.

  1. 222 – might you consider further study for Bethanechol as recently suggested – 50 mg/BID needs further testing! Walter, J.; Wheeler, J. Use of Bethanechol, 50 mg/BID, for a Failed Decatheterization Test: A Position Statement. Uro 20222, 134–136.   https://doi.org/10.3390/uro2020016

Reply: We thank the reviewer for the important hint. We have added the suggested reference to our article.

  1. However, very expensive

Reply: We thank the reviewer for the valued comment. We added the aspect of higher therapy cost to the section.

  1. Please add a section on pessary use and outcomes – this should also be first-line therapy

Reply: We thank the reviewer for the important comment. We added a section on pessary use and outcomes, as it is an important treatment of SUI.